# Data Valuation in the Absence of a Reliable Validation Set

**Himanshu Jahagirdar**                                                          *himanshugj@vt.edu*
*Virginia Tech*

**Jiachen T. Wang**                                                          *tianhaowang@princeton.edu*
*Princeton University*

**Ruoxi Jia**                                                          *ruoxijia@vt.edu*
*Virginia Tech*

**Reviewed on OpenReview:** https://openreview.net/forum?id=xBORyL316c

## Abstract

Data valuation plays a pivotal role in ensuring data quality and equitably compensating data contributors. Existing game-theoretic data valuation techniques mostly rely on the availability of a high-quality validation set for their efficacy. However, the feasibility of obtaining a clean validation set drawn from the test distribution may be limited in practice. In this work, we show that the choice of validation set can significantly impact the final data value scores. In order to mitigate this, we introduce a general paradigm that converts a traditional validation-based game-theoretic data valuation method into a validation-free alternative. Specifically, we utilize the cross-validation error as a surrogate for to evaluate the model's performance on a validation set. As computing the cross-validation error can be computationally expensive, we propose using the cross-validation error of a kernel regression model as an effective and efficient surrogate for the true performance score on the population. We compare the performance of the validation-free variant of existing data valuation techniques with their original validation-based counterparts. Our results indicate that the validation-free variants generally match or often significantly surpass the performance of their validation-based counterparts.

## 1 Introduction

Data valuation aims to measure the contribution of individual data instances to the training of machine learning models. The task of data valuation is crucial not only in data marketplaces where it ensures equitable compensation for data providers, but also in the realm of explainable machine learning - where it identifies influential training data points that are responsible for certain model behavior. The importance of data valuation research is underlined by legal efforts such as the DASHBOARD Act (Foster, 2019) and Data Dividend Project (Project, 2020), mandating companies to assess the economic value of user data.

The Shapley value is a well-known solution concept in cooperative game theory (CGT) that fairly divides total revenue among players. Originally proposed by Jia et al. (2019b) and Ghorbani & Zou (2019), the Shapley value has become one of the most popular approaches for data valuation. Specifically, each training data point is considered a "player" in a coalitional game, and the Shapley value is used to fairly evaluate the contribution of each player. The technique is often termed *"Data Shapley"*. Other CGT-based approaches have been developed, including Beta Shapley (Kwon & Zou, 2021), Data Banzhaf (Wang & Jia, 2023a), etc. The primary reason for the success of Data Shapley and the other CGT-based approaches is their axiomatic formulations, which intuitively resonate with the fairness requirements inherent in data valuation. They have been adopted in various use cases such as improving dataset quality (Tang et al., 2021; Pandl et al., 2021), incentive mechanism design (Liu et al., 2020; Wei et al., 2020), data debugging (Karlaš et al., 2022).

A key issue that is often overlooked in data valuation research is the dependency on a well-curated validation data. The choice of validation data can often greatly impact the outcome of data valuation, potentially

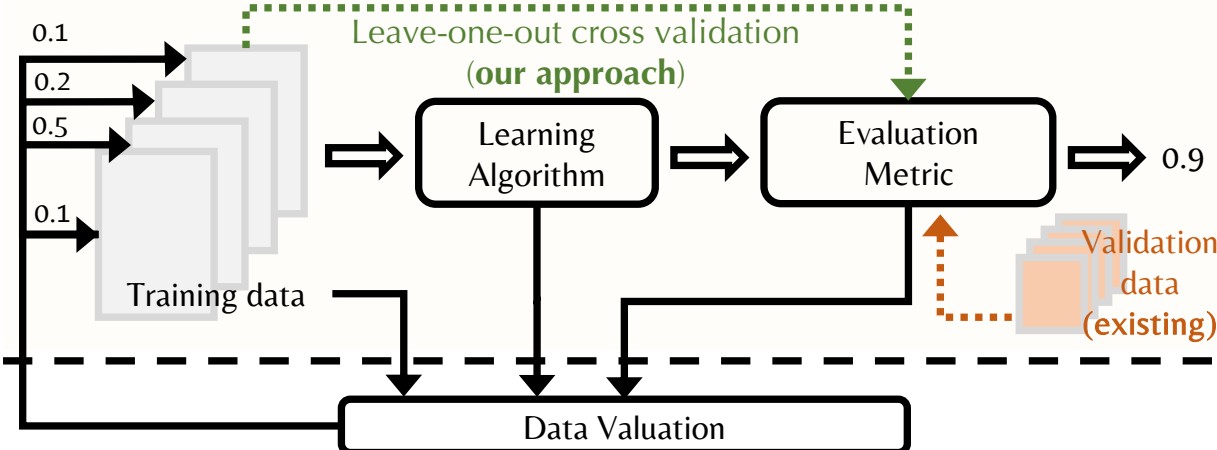

Figure 1: Overview of our proposed approach vs. traditional data valuation approaches.

leading to varying results. Moreover, procuring a clean, representative validation set is often a challenging task, and in many real-world scenarios, the size of the available validation set is limited. In addition, data valuation's sensitive nature, specifically concerning profit sharing, introduces additional complexities. If certain individuals possess insider knowledge about the validation set, they might strategically optimize their contribution to the training data. This could potentially skew the profit distribution in their favor, thereby introducing unfairness.

In this paper, we uncover the risks of using validation data for data valuation, and propose a general paradigm that converts the many existing validation-dependent data valuation approaches into validation-free approaches. Our technical contributions are listed as follows:

**Uncovering the potential vulnerabilities of validation-based data valuation techniques.** Through a series of experiments, we demonstrate that the choice and quality of the validation set can significantly affect data values. The size of the validation set directly impacts the ranking of these values. Moreover, we show that even minor class-imbalance within the validation set can substantially alter data values. Finally, we highlight that these valuation systems can be gamed and become unfair if a malicious data provider gains access to information about the validation set. Our findings underscore the need for caution when using these techniques, due to the potential for significant variations and vulnerabilities.

**A validation-free paradigm for data valuation using Leave-One-Out Cross-Validation (LOOCV).** Recognizing the limitations of validation-based data valuation techniques, we propose a novel validation-free approach using Leave-One-Out Cross-Validation (LOOCV) to estimate performance scores on the population. Cross-Validation (CV) is a classic technique for estimating model performance scores when a validation set is unavailable. LOOCV's deterministic nature makes it an ideal alternative utility function in data valuation. To reduce the computational demands associated with LOOCV, we propose to use Regularized Least Squares (RLS) as an efficient proxy model tailored for validation-free data valuation. This is due to the special property of RLS in having a computational shortcut for LOOCV. While proxy models are frequently being used to overcome computational challenges in data valuation (Jia et al., 2019a; Kwon & Zou, 2023), the existing proxy models yield limited advantages in our context, primarily because their associated LOOCV computation is still inefficient. Our novelty lies in developing an efficient validation-free data valuation algorithm based on appropriate proxy models. We complement our method with an error analysis for using LOOCV as utility scores. Figure 1 shows the overview of the proposed paradigm.

**Empirical Evaluation.** We demonstrate the effectiveness of LOOCV-based data valuation techniques on important downstream tasks. Compared with validation-based techniques, we show that LOOCV-based data valuation techniques achieve comparable performance on the weighted accuracy task and (often) superior performance on noisy label detection task. We also show that RLS with Gaussian kernel as a proxy model is an effective proxy model for valuation, the computed data value scores have a better performance on these

downstream tasks than the validation-based counterparts. We explain this behavior using the locality-aware nature of Gaussian kernels that we use in our RLS implementation.

Overall, our work provides a robust solution to the challenges posed by validation-set dependencies in data valuation, and takes one step further towards a more fair and practical approach to assessing data value.

## 2 Background

In this section, we introduce the problem and setup of data valuation in the context of machine learning, and we provide a brief overview of the Shapley value and other related work.

**Valuing Data for Machine Learning Applications.** Consider a dataset $D = \{z_i\}_{i=1}^N$ where each $z_i = (x_i, y_i) \in (\mathcal{X} \times \mathcal{Y})$ for data space $\mathcal{X}$ and label space $\mathcal{Y}$. The aim of data valuation is to assign an importance score to each data point in a dataset, reflecting its value in training the model. The "contribution" of a data point is typically assessed based on a *utility function* $U : \cup_{n=0}^\infty (\mathcal{X} \times \mathcal{Y})^n \to \mathbb{R}$. This function assigns a score to any set of training data points, indicating its utility. Ideally, one would like to select $U$ as the test performance of the model trained on a given dataset and evaluated on the population distribution $\mathcal{P}$. However, in practical scenarios, the population might not be known at the time of training. Consequently, model performance is commonly assessed on a validation set $D_{\text{val}}$ that is drawn from $\mathcal{P}$. Formally, for any dataset $S$, we define $U_{D_{\text{val}}}(S) := \texttt{Perf}_{D_{\text{val}}}(\mathcal{A}(S))$, where $\mathcal{A}$ is a learning algorithm that takes dataset $S$ as input and returns model trained on it, and $\texttt{Perf}_{D_{\text{val}}}$ denotes the validation performance of the given model evaluated on the validation set $D_{\text{val}}$. We also denote $U_{\mathcal{P}}(S) := \texttt{Perf}_{\mathcal{P}}(\mathcal{A}(S))$ as the model performance evaluated on the population distribution. Given a dataset $D = \{z_i\}_{i=1}^N$, the ultimate objective of data valuation is to compute a score vector $(\phi_{z_i}(U_{\mathcal{P}}))_{z_i \in D}$, where each $\phi_{z_i}(U_{\mathcal{P}})$ signifies the importance of data point $z_i$ to the resulting model performance evaluated on the population distribution. In practice, however, one usually use $U_{D_{\text{val}}}$ to approximate $U_{\mathcal{P}}$ and thus compute the score vector as $(\phi_{z_i}(U_{D_{\text{val}}}))_{z_i \in D}$.

**Data Shapley.** The Shapley value (Shapley, 1953) is a classic concept in cooperative game theory to attribute the total gains generated by the coalition of all players. At a high level, it appraises each point based on the (weighted) average utility change caused by adding the point into different subsets. Given a utility function $U(\cdot)$ and a dataset $D = \{z_i\}_{i=1}^N$, the Shapley value of a data point $z_i \in D$ is defined as

$$\phi_{z_i}^{\text{shap}}(U) := \frac{1}{N} \sum_{k=1}^N \binom{N-1}{k-1}^{-1} \sum_{S \subseteq D \setminus \{z_i\}, |S| = k-1} [U(S \cup \{i\}) - U(S)]$$

The popularity of the Shapley value is attributable to the fact that it is the *unique* data value notion satisfying certain reasonable axioms (Shapley, 1953). We refer the readers to Ghorbani & Zou (2019); Jia et al. (2019b) and the references therein for the description and discussion about these axioms in the ML context.

**Data Banzhaf.** The Banzhaf value (Banzhaf III, 1964) is another classic concept in cooperative game theory to attribute the total gains to individual players. The Banzhaf value averages the marginal contribution across all subsets. Formally, given a utility function $U(\cdot)$ and a dataset $D = \{z_i\}_{i=1}^N$, the Banzhaf value of a data point $z_i \in D$ is defined as

$$\phi_{z_i}^{\text{banz}}(U) := \frac{1}{2^{N-1}} \sum_{S \subseteq D \setminus \{z_i\}} [U(S \cup \{i\}) - U(S)]$$

Recently, Wang & Jia (2023a) discovered that the Banzhaf value is the most robust data value notion among the class of semivalues (Dubey et al., 1981), and thus is suitable for data valuation especially when the underlying learning algorithm is stochastic.

**Related work.** Multiple recent works have been carried out to relax the assumptions and stringent requirements of the Data Shapley framework. Wu et al. (2022); Just et al. (2023); Nohyun et al. (2022); Jia et al. (2019a); Kwon & Zou (2023); Yoon et al. (2020) propose different data valuation paradigms that do not require training neural networks multiple times, sidestepping the often cumbersome model re-training processes. For instance, methods like KNN-Shapley (Jia et al., 2019a) and Data-OOB (Kwon & Zou, 2023)

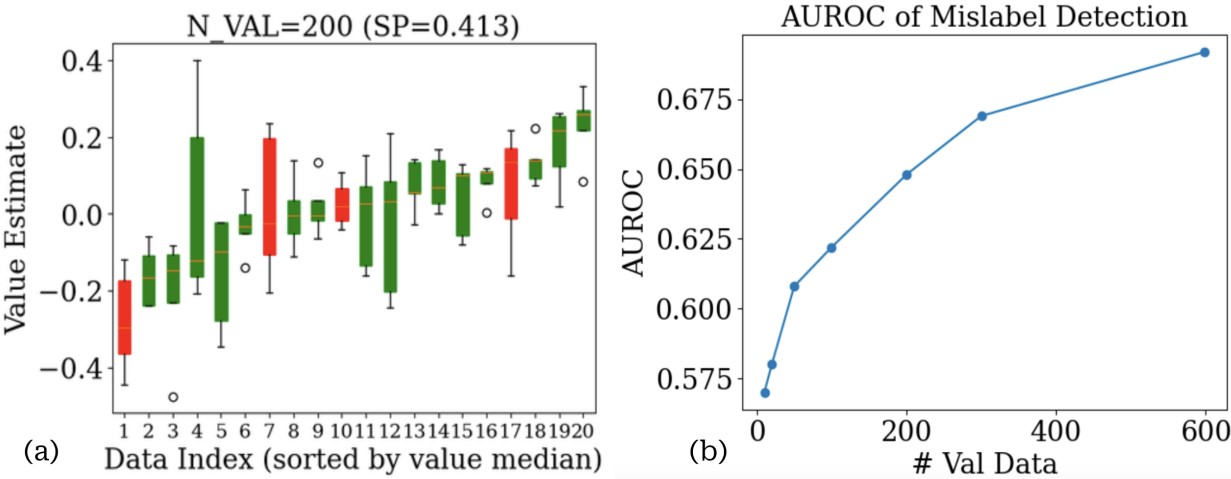

Figure 2: (a) Box plot depicts the data values(using Data Shapley) for 20 points randomly sampled from the Census Dataset over 5 different validation sets of equal size. The mislabeled points are in red. (b) The performance on noisy label detection (quantified by AUROC scores) with different validation set sizes.

use smaller proxy models as the surrogate for the original learning algorithms, which effectively optimize the computational efficiency of the valuation process. While being computationally efficient, this line of work requires a clean validation set that is drawn from the target distribution. In contrast, our research introduces a novel proxy model tailored for an efficient extension to a validation-free setting. Another line of works Kwon & Zou (2021); Wang & Jia (2023a); Lin et al. (2022); Yan & Procaccia (2020) have sought to relax certain axioms intrinsic to the Shapley value, and these techniques turn out to outperform Data Shapley in certain aspects such as the reproducibility. Xu et al. (2021) also aim at removing the assumption of the availability of a clean, representative validation set for data valuation, and are the most relevant to our study. It leverages the diversity of the dataset (quantified by the determinant of the data matrix) to assess data value. However, this approach harbors a notable drawback that it does not incorporate label information. Hence, this technique will be ineffective for crucial applications such as detecting mislabeled data. Data-OOB (Kwon & Zou, 2023) is particularly good at detecting mislabeled data, and also does incorporates label information , making it relevant to our work. It is important to note that our method differs in the ability to convert existing game-theoretic valuation frameworks into a validation-free one, while these methods do not satisfy the axioms that Shapley-based values or semi-values do.

## 3 The Achilles' heel of Validation-Based Data Valuation

Validation-based data valuation approaches can provide fair and intrinsic value given access to a clean, unbiased, and relatively large validation set. However, this validation set needs to be sourced independently from the training data, which is often difficult in practice. A common strategy to deal with this issue is to take a piece of training set as the hold-out validation set. However, this can cause a large variance in the final performance score due to the randomness in the partition. Moreover, it is hard to ensure all edge cases are fairly represented in this validation set. If the validation set is shifted from the population distribution, certain training points may unfairly get higher or lower values. Procurement of the validation data is not the only challenge. Even when a clean and unbiased validation set is available, the choice of validation set can be a big factor in determining the values obtained. Preparation of such a validation set for valuation can become a bottleneck. In this section, we highlight situations when the choice of validation set affects the nature of values obtained. Specifically, we study the effect of **(1) Choice**, **(2) Size** and **(3) Class Distribution** of the underlying validation data on data valuation. We also study the effects of **gaming** the valuation framework given access to validation data. For all experiments, we consider multiple datasets, presenting one in the main paper and deferring the rest to Appendix B.2 due to the similar nature of results.

**Choice of Validation Set.** This experiment tries to show how the choice of a fixed-size validation set can affect the quality of data values obtained. We attempt to value a 1000-sized subset of the Census Dataset

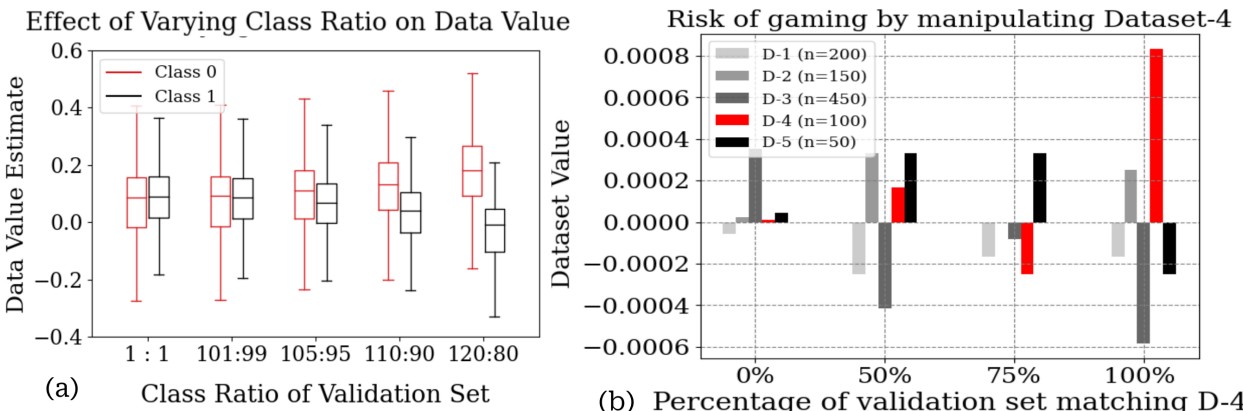

Figure 3: (a) Box plots show how data values for samples are influenced by the class distribution of the underlying validation set. (b) Risk of gaming dataset valuation is shown by choosing a validation set that matches Dataset 4.

from the UCI Repository (Dua & Graff, 2017). We randomly sample the test set without replacement to obtain 5 distinct 200-sized validation sets, drawn from the same distribution. We examine two popular existing data valuation frameworks, the Data Shapley (Ghorbani & Zou, 2019) and KNN-Shapley (Jia et al., 2019a). Data Shapley is the most famous model-dependent data valuation technique, and KNN-Shapley has been recognized as the most practical training-free data valuation technique. From Fig. 2 (a) and Appendix B.2, we observe that values vary, sometimes significantly, depending on the validation set used, despite the validation sets being drawn from the same distribution. Many points are valued positively under one validation set, and negatively under another. We also observe that clean and mislabeled points are equally affected by the choice of validation data. The variance of their values is substantial enough that the order of their values varies based on different validation sets. Thus, the typical mislabeled detection task by removing the lowest-valued points can perform differently with different sampled validation sets. While it is unsurprising that the value depends on the validation choice, the fact that even independently drawn validation sets from the same distribution significantly affect the order of data values is concerning. The reason for this instability of ranking is that the data values for individual points differ only by a small magnitude.

**Size of Validation Set.** For this experiment, we take a 1000-size subset of the Credit Card Data (Yeh & Lien, 2009) and vary the size of validation set. We study the performance of data values in a mislabeled detection task. We use KNN-Shapley for valuation since it performs well for this dataset. It is expected that with more validation data, we get better detection rates as we can see from Fig. 2 (c). Choosing the size of validation set depends on the availability of data. When validation data is not available in abundance, obtained values may not be optimal.

**Class-imbalance in Validation Set.** In this setup, we assume a balanced training class distribution and a varying validation class distribution. We demonstrate how sensitive data values are to small changes in class ratio - and how that can be harmful for valuation. We perform this experiment on a 200-sized Census dataset and vary the class ratio in the validation set, using Data Shapley as the choice of valuation framework. As we can see from Fig. 3 (a), for a size 200 validation set, with a realistic class ratio of 105:95 we see the majority class samples getting higher values on average. This problem is magnified if the ratio is further increased. The true class distribution of the Census data is actually quite skewed (since most people have income below 50K). The dilemma on whether to remain true to the actual distribution or attempt a balanced validation set (creating feasibility issues). The takeaway here is that choice of validation set is not simple and design choices play a significant part in values obtained.

**Risk of Gaming a Data Valuation System.** An important task of data valuation is to attribute equitable payoff to multiple data sources in collaborative learning (Jia et al., 2019b). Fig. 3(b) shows a toy experiment suggesting how manipulation of validation data can game the valuation framework. We have 5 data sources (D1-D5), each with varying data size - randomly split from the Census dataset. The valuation framework

used is Data Shapley- since it can be easily extended to dataset valuation. To simulate gaming the validation set, we have selected a percentage of D-4 (in red) to match the validation set. When the validation data is independently sourced (0% match with D-4), we see that dataset D-3 (which contributes the largest number of data points) is rightly attributed the highest value. However, when 50% of the validation set matches D-4, we see that D-3 has a very low value (suggesting that it should be discarded). Moreover, D-2 and D-5 suddenly see a sharp increase in importance and are likely to be selected over D-3. When a higher percentage (75% and 100%) of D-4 matches with the validation set, we see that D-4 has a very high data value.

This experiment is a naive effort to show that if a malicious data provider has information about the validation set, they can manipulate their data to ensure selection, which can adversely affect the expected payoff to other providers. When validation data is limited and not necessarily private, it can risk gaming the valuation framework.

## 4 Proposed Approach

To address the challenges associated with the choice of validation data for data valuation, we propose a general paradigm that leverages *cross-validation* (CV), a widely adopted method in statistical machine learning. CV is renowned for its ability to provide reliable estimates of model performance, even in the absence of a dedicated validation set. By substituting the traditional validation accuracy with the CV error, we eliminate the necessity for a clean, representative validation set. This effectively **transforms the conventional validation-based game-theoretic data valuation framework into a validation-free alternative**. By removing this dependence on the choice of the validation set, we avoid the aforementioned issues associated with the quality, size, and bias of the validation set. Moreover, it reduces the risks of having malicious players gaming with the validation set.

### 4.1 Cross Validation as Utility Function

Cross-validation (CV) is a widely-used technique in statistical machine learning for estimating the generalizability of a trained model to the population distribution. In a $K$-fold CV, data is randomly partitioned into $K$ equal-sized subsets. The model is trained on $K-1$ subsets and tested on the remaining one, repeating this process $K$ times and averaging the validation performance over the remaining subset. Leave-one-out cross-validation (LOOCV) is a special case of $K$-fold where $K$ equals the total sample size. That is, it trains the model on all data points except one, and repeats this for each data point. Compared with other $K$-fold CV, LOOCV is often preferred due to its deterministic nature and the elimination of the need to tune $K$.

Given LOOCV's ability to provide reliable estimates of model performance without the need for validation data, it emerges as a natural choice for use as an alternative utility function in the context of data valuation without a dedicated validation set. Formally, for a pre-specified learning algorithm $\mathcal{A}$, we define the utility function that uses LOOCV as $U_{\texttt{LOOCV}}(S) := \frac{1}{|S|} \sum_{i \in S} \texttt{Perf}_{z_i}(\mathcal{A}(\{z_j\}_{j \in S \setminus i}))$.

**Computational Challenge of using LOOCV as Utility Functions.** The challenge of using (3) as the utility function is the substantial computational overhead that the calculation of LOOCV introduces. While validation-based utility functions $U_{D_{\text{val}}}(S)$ require training a single model on the input dataset $S$, each evaluation of $U_{\texttt{LOOCV}}(S)$ necessitates training a model for each subset of $S \setminus z_i$, for each $i \in S$. This is computationally prohibitive for modern learning algorithms such as neural networks.

### 4.2 Efficient LOOCV Computation for Regularized Least Square (RLS)

**Proxy Model Approach for Data Valuation.** A prevalent strategy in data valuation for improving computational efficiency is to utilize proxy models that allow efficient evaluation of utility functions. A prominent example is the KNN-Shapley (Jia et al., 2019a), which uses K-Nearest Neighbors (KNN) as a proxy model with a closed-form solution for exact Shapley value computation. While justifying the use of KNN as a proxy model for data valuation on theoretical grounds remains challenging, KNN-Shapley has been recognized as one of the most practical data valuation techniques due to its computational efficiency and effectiveness in distinguishing data quality. Drawing inspiration from this, we turn to Regularized Least Squares (RLS) as our proxy model, given its unique property of having an efficient formula for computing LOOCV (Pahikkala et al., 2006).

**Regularized Least Squares (RLS).** Formally, given a dataset $D = \{(x_1, y_1), \ldots, (x_N, y_N)\} \in (\mathcal{X} \times \mathbb{R}^p)^N$ and a kernel function $k(\cdot, \cdot)$, a regularized least squares kernel regression seeks to minimize the following objective:

$$\min_f \sum_{i=1}^{N} (y_i - f(x_i))^2 + \lambda \|f\|_k^2 \tag{1}$$

where $f : \mathcal{X} \to \mathbb{R}^p$, $\lambda > 0$ is a regularization coefficient, and $\|\cdot\|_k$ is a norm in a Reproducing Kernel Hilbert Space (RKHS) associated with the kernel $k(\cdot, \cdot)$. Let $K \in \mathbb{R}^{N \times N}$ be the kernel matrix with $K_{ij} = k(x_i, x_j)$, and let $Y = [y_1, \ldots, y_N]^T$. Let $G := (K + \lambda I_N)^{-1}$ where $I_N$ is the identity matrix. Because the kernel function from which the kernel matrix is generated is positive definite, the matrix $K + \lambda I_N$ is invertible when $\lambda > 0$. The solution to this optimization problem can be expressed as $f(x) = k(x)^T G Y$ where $k(x) := [k(x, x_1), \ldots, k(x, x_N)]^T$.

**Efficient Computation of LOOCV for RLS.** One of the well-known results of RLS is its short-cut formula for computing the leave-one-out model directly from the model trained on the full dataset. This result leads to an efficient computation method for LOOCV. Specifically, denote the matrix $A := KG$, and let $a_i := [I_i K G I_i^T]$ be the $i$-th diagonal element of matrix $A$, where $I_i$ is the i-th row of identity matrix $I_N$. At a high level, the shortcut for RLS's LOOCV formula exists because of the combined effects of the linear nature of RLS's predictions and the structure provided by the hat matrix, which together allows for efficiently computing the change in predictions when a training data point is omitted and being evaluated on.

**Theorem 1** (Pahikkala et al. (2006)). *Let $f$ denote the model of RLS trained on dataset $S$, and $f_{-i}$ denote the model of RLS trained on the leave-one-out dataset $S \setminus (x_i, y_i)$. Then we have*

$$f_{-i}(x) = \frac{f(x_i) - a_i y_i}{1 - a_i} \tag{2}$$

Leveraging this efficient computation of the leave-one-out model, we can compute the LOOCV score for RLS with any utility metric. The main advantage we gain by using this LOOCV linear formulation is that we can evaluate the utility of a set once and efficiently obtain the leave-one-out models (and the marginal contributions) without the need for re-training. Hence, we eliminate the additional matrix inversion operations that are otherwise necessary to train these leave-one-out models-making our method a desirable candidate for many data valuation frameworks.

**Corollary 2** (LOOCV Formula for RLS for arbitrary utility metric (Pahikkala et al., 2006)). *Let $f$ denote the model of RLS trained on dataset $S$. The LOOCV error for RLS on dataset $S$ can be computed using the following formula:*

$$U_{LOOCV}(S) := \frac{1}{|S|} \sum_{i \in S} \textit{Perf}_{z_i}(f_{-i}) \tag{3}$$

*where $f_{-i}$ can be efficiently computed via (2).*

This corollary allows us to efficiently compute the LOOCV error, which is a key component of our proposed utility function for data valuation. Additionally, it opens the potential for parallel computation of $f_{-i}$ for all $i$ via GPU operations, though we do not explore this setup in this work. This utility metric can be instantiated for both regression and classification tasks by choosing the appropriate (task-specific) performance metric function `Perf`. Extending $f$ to classification tasks further involves a one-hot style label encoding using $\pm 1$.

**Remark 3** (Computational efficiency). *When using $U_{LOOCV}$ as the alternative utility function for computing Data Shapley and other CGT-based values (such as Data Banzhaf (Wang & Jia, 2023a)), the computational overhead with respect to the number of models trained (that is, the number of linear regressions to fit) remains identical to the case when using regular utility functions. We refer the readers to Wang & Jia (2023b;a) for the detailed sample complexity results for different types of Monte Carlo estimators. Additionally, once the regression model is fit, the computation of LOOCV only requires model predictions, an operation which, in terms of computational demand, is negligible compared with model fitting. In Appendix B.7, we discuss the FLOPS analysis and runtime advantage for $U_{LOOCV}$ when compared to its re-training based alternatives.*

### 4.3 Error Analysis

Recall that the objective in data valuation is to compute the data value score $\phi_{z_i}(U_{\mathcal{P}})$, where $U_{\mathcal{P}}(\cdot)$ is the utility function defined by the model's performance evaluated on the population distribution $\mathcal{P}$. In this section, we provide an error analysis for the use of $U_{D_{\mathrm{val}}}$ and $U_{\mathrm{LOOCV}}$ as replacements for $U_{\mathcal{P}}$ in computing the celebrated Shapley value, specifically for the case of linear regression models. That is, for a random dataset $D$ drawn from the population distribution $\mathcal{P}$, we analyze the expected deviation from $\phi_{z_1}(U_{\mathcal{P}})$ when using $U_{D_{\mathrm{val}}}$ or $U_{\mathrm{LOOCV}}$ to compute the Shapley value of a data point $z_1 \in D$. We use $\phi_{z_1}(U_{\mathcal{P}}; D)$ to stress the underlying dataset for computing the data value score.

Our first result shows that, if the validation set $D_{\mathrm{val}}$ is sampled from $\mathcal{P}$ and has a size of $k$, then the deviation $|\phi_{z_1}(U_{D_{\mathrm{val}}}) - \phi_{z_1}(U_{\mathcal{P}})|$ is upper bounded by $O(1/\sqrt{k})$ on expectation.

**Theorem 4** (Expected Error Bound for $\phi_{z_1}(U_{D_{\mathrm{val}}})$). *If both $U_{\mathcal{P}}$ and $U_{D_{\mathrm{val}}}$ have bounded range, then we have*

$$\mathbb{E}_{D \sim \mathcal{P}^N, D_{\mathrm{val}} \sim \mathcal{P}^k} \left[ |\phi_{z_1}(U_{D_{\mathrm{val}}}; D) - \phi_{z_1}(U_{\mathcal{P}}; D)| \right] \leq O\left( \frac{1}{\sqrt{k}} \right)$$

Theorem 4 is based on the fact that if $U_{\mathcal{P}}$ are bounded, then for any training set $S$, we have $\mathbb{E}_{D_{\mathrm{val}} \sim \mathcal{P}^k} \left[ |U_{\mathcal{P}}(S) - U_{D_{\mathrm{val}}}(S)| \right] \leq O\left( 1/\sqrt{k} \right)$ which is the standard sample error bound.

Next, we show that at least for the case when the utility means the (negative of) mean squared loss of RLS, $\phi_{z_1}(U_{\mathrm{LOOCV}})$ can achieve a similar guarantee.

**Theorem 5** (Expected Error Bound for $\phi_{z_1}(U_{\mathrm{LOOCV}})$). *If both $U_{\mathcal{P}}$ and $U_{\mathrm{LOOCV}}$ are average absolute prediction error and have a bounded range, then for linear regression models, we have*

$$\mathbb{E}_{D \sim \mathcal{P}^N} \left[ |\phi_{z_1}(U_{\mathrm{LOOCV}}; D) - \phi_{z_1}(U_{\mathcal{P}}; D)| \right] \leq O\left( \frac{1}{\sqrt{N}} \right)$$

Theorem 5 is based on the previous result for the LOOCV guarantee $\mathbb{E}_{D \sim \mathcal{P}^N} \left[ |U_{\mathcal{P}}(S) - U_{\mathrm{LOOCV}}(S)| \right] \leq O\left( 1/\sqrt{N} \right)$ for linear regression models (Tian et al., 2007). It tells us that, at least for the case of linear regression, when the size of available validation data $k$ is significantly less than the training set size $N$, the Shapley value derived from $U_{\mathrm{LOOCV}}$ can be closer to the ground truth compared with the Shapley value derived from $U_{D_{\mathrm{val}}}$.

**Remark 6** (**Further discussion about Theorem 5**). *(1) Applicability. Although Theorem 5 is stated specifically for the case of linear regression, this bound is also applicable to other learning algorithms where there have been previous results on the error bound of LOOCV, such as Ridge regularized logistic regression (Rad et al., 2020). (2) Error of using RLS as proxy model. It is important to note that our error analysis does not account for using RLS as a proxy model for the primary learning algorithm. The closeness between the utility scores from different learning algorithms, however, is generally challenging to analyze and has been noted in the literature (Coleman et al., 2019). (3) Error of sampling from an alternative distribution. In our error analysis for Theorem 5, we have assumed that the training data $D$ is drawn from $\mathcal{P}$. However, in practical scenarios, there may be a distribution shift, i.e., $D$ may be sampled from an alternative distribution $\mathcal{P}'$. This scenario can be analyzed using theoretical results from domain adaptation, and such a distribution will introduce an additional error term characterized by the distance between $\mathcal{P}$ and $\mathcal{P}'$, which is independent of the size of the training data. See Appendix A for details.*

## 5 Evaluation

We have evaluated our method to answer the following questions: **(1)** Can our Kernel Regression-based LOOCV method transform existing game-theoretic (validation-based) valuation techniques into a validation-free framework, while retaining performance on standard valuation tasks? **(2)** How does our method compare to existing validation-free approaches or naive solutions? If our method performs at par with existing validation methods, we also offer insight into why Kernel Regression is a suitable candidate for valuation.

| | Data Banzhaf | | | Data Shapley | | |
|---|---|---|---|---|---|---|
| Dataset | LOOCV | Self Eval | Val Based | LOOCV | Self Eval | Val Based |
| CIFAR10 | **0.6992** | 0.5088 | 0.5872 | **0.5664** | 0.5216 | 0.574 |
| MNIST | **0.965** | 0.6419 | 0.7209 | **0.8507** | 0.8426 | 0.87 |
| Census | **0.883** | 0.58 | 0.587 | **0.803** | 0.668 | 0.595 |
| Phoneme | **0.862** | 0.5178 | 0.537 | 0.739 | **0.7396** | 0.7269 |
| Apsfail | **0.8905** | 0.8762 | 0.7711 | **0.8042** | 0.7752 | 0.8458 |
| cpu | **0.9233** | 0.9016 | 0.846 | **0.915** | 0.8969 | 0.9233 |
| Fraud | 0.9694 | **0.9808** | 0.883 | 0.9735 | **0.9838** | 0.9855 |
| Pol | **0.8969** | 0.3922 | 0.9330 | **0.8969** | 0.6734 | 0.9583 |
| vehicle | **0.6983** | 0.6896 | 0.753 | **0.6747** | 0.6668 | 0.7469 |

Table 1: AUROC score of mislabeled points detection using different data valuation techniques on 9 datasets. We highlight the best results of *validation-free approaches*.

| | Data Banzhaf | | | Data Shapley | | |
|---|---|---|---|---|---|---|
| Dataset | LOOCV | Self Eval | Val Based | LOOCV | Self Eval | Val Based |
| CIFAR10 | **0.7375** | 0.6915 | 0.742 | **0.7275** | 0.704 | 0.739 |
| MNIST | **0.7986** | 0.785 | 0.803 | **0.7992** | 0.7899 | 0.76 |
| Census | 0.806 | **0.8179** | 0.7915 | **0.812** | 0.8017 | 0.805 |
| Phoneme | **0.7635** | 0.7625 | 0.7647 | **0.76** | **0.76** | 0.7618 |
| Apsfail | **0.9115** | 0.9098 | 0.9086 | **0.9076** | 0.9066 | 0.9073 |
| cpu | **0.8899** | 0.8885 | 0.8935 | **0.8903** | 0.888 | 0.8905 |
| Fraud | 0.9258 | **0.929** | 0.931 | 0.923 | **0.928** | 0.926 |
| Pol | **0.8356** | 0.835 | 0.859 | 0.836 | **0.84** | 0.847 |
| vehicle | 0.807 | **0.815** | 0.816 | 0.806 | **0.816** | 0.8164 |

Table 2: Accuracy comparison of models trained with samples weighted by different data valuation techniques. We highlight the best results of *validation-free approaches*.

## 5.1 Experimental Setup

We summarize important experiment settings here, and additional details are available in Appendix B.

**Data valuation frameworks.** We apply LOOCV to existing CGT-based data valuation frameworks to render them validation-independent. Our goal is to understand how this process affects the usefulness of the derived data values. We consider two commonly used CGT-based frameworks - Data Shapley (Ghorbani & Zou, 2019) and Data Banzhaf (Wang & Jia, 2023a). We use the state-of-the-art approximation algorithms for these two frameworks (see Appendix B for details). We also include a comparison with Beta Shapley (Kwon & Zou, 2021) in Appendix B.3. Our method is not compatible with KNN-Shapley ( a special instance of Data Shapley), hence we do not test it with our method.

**Implementation details.** We evaluate data values over 9 classification datasets popularly used in data valuation literature (refer Appendix B.1). For each comparison study, validation-based baselines use standard models (either binary MLP or logistic regression) that initially perform the best on a select held-out validation set. LOOCV calculation (outlined in Theorem 2) involves computing the efficient cross-validation accuracy (using Theorem 5) on an RLS model ($\lambda = 0.1$) with a Gaussian Kernel. Additionally, we perform an ablation study on the effect of changing parameter $\lambda$ in Appendix B.6.

**Baselines.** A *novel baseline* that we explore is the usage of the whole training set as a substitute for the validation set. This naive approach (henceforth mentioned as **Self-Eval**) helps us verify the superior performance that cross-validation can provide over simply using the whole training dataset as the validation set for attribution. The Self-Eval baseline also uses the RLS model. Our second baseline is the validation-dependent version of the same data valuation framework, where we set the size of the validation set the same as the size of the training set. Note that this baseline assumes more knowledge than our approach and Self-Eval and thus their results are not directly comparable. Nevertheless, having this baseline helps us to evaluate whether game-theoretic validation-free valuation approaches can yield results as competitive as those from validation-based approaches.

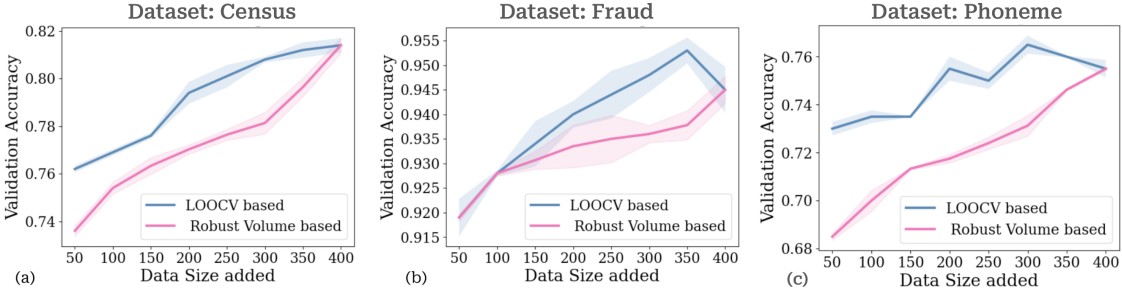

Figure 5: Comparison between LOOCV and Volume (Xu et al., 2021) on a dataset addition experiment (highest valued dataset added first) with (a)- Census Dataset, (b) Fraud Dataset and (c) Phoneme Dataset.

## 5.2 Quality of Values Obtained

We study the quality of data values obtained using two different criteria- the first is performance (AUROC) on the noisy label detection task. In all experiments, labels have been randomly flipped with a fixed poison ratio of 10%. A higher AUC of the detection rate suggests better detection performance, which in turn supports the quality of values obtained. Our second criterion to assess quality is learning with weighted samples. Comparison with a validation-based baseline will test if LOOCV can provide equally useful data values. In comparing with Self-Eval, we test how different designs of validation-free schemes affect utility of derived values.

**Mislabeled Data Detection.** Table 1 shows the results for mislabeled detection. We found that LOOCV can get comparable and often superior performance from traditional validation-based approaches. Noisy label detection is quite challenging and the strong performance reinforces that our data values can compete with validation-based counterparts.

Fig. 4 shows how the AUROC varied when we increased the dataset size for the OpenML Phoneme speech-recognition dataset. We see that for this dataset, LOOCV can achieve high performance for larger sizes but self-eval performance noticeably drops.

We perform an ablation study varying the mislabeled ratio and evaluating AUROC scores on LOOCV, Self-Eval, and Validation-based methods in Appendix B.6. As per intuition, we find that there is a slight decrease in AUROC scores as the noisy label ratio increases to 20%- since our method relies on training data for evaluation. However, our AUROC scores were always higher than those obtained from validation-based counterparts. Additional evaluation (using Beta Shapley framework in Table 5) and settings can be found in Appendix B.

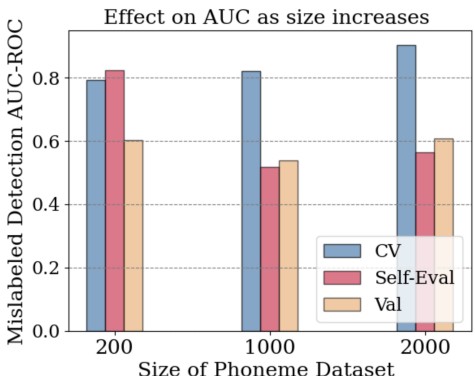

Figure 4: Comparison of AUROC on mislabeled detection task with different data valuation techniques as the size of the dataset increases.

**Weighted Accuracy.** We further conduct a weighted training experiment in Table 2. We weigh the dataset with the data value obtained from Data Shapley and Data Banzhaf. After obtaining data values, we test the performance on an independent test set that was not used during valuation. We observe that test accuracies obtained using LOOCV closely track their validation-based counterparts.

## 5.3 Comparison with Existing Validation-free Frameworks

**Comparison with Volume-based data valuation (Xu et al., 2021).** In Appendix B.4 we conduct a comparison with Volume (Xu et al., 2021), a label-agnostic and validation-free valuation method. We find that it is unable to perform well on mislabeled-detection tasks (due to its label-agnostic nature) and our weighted accuracies usually perform better than Volume on the standard datasets. We note that Volume is designed for Dataset Valuation and we conduct an experiment for comparing LOOCV and Volume in that

| Dataset | AUCROC LOOCV | AUCROC Data-OOB | Weighted Acc. LOOCV | Weighted Acc. Data-OOB |
|---------|--------------|-----------------|---------------------|------------------------|
| Census  | 0.8830 | 0.9429 | 0.8060 | 0.7533 |
| Phoneme | 0.8620 | 0.8992 | 0.7635 | 0.7542 |
| Apsfail | 0.891  | 0.976  | 0.9115 | 0.9232 |
| CPU     | 0.9233 | 0.9362 | 0.8899 | 0.8792 |
| Fraud   | 0.9694 | 0.9814 | 0.9258 | 0.8899 |
| Pol     | 0.8969 | 0.9486 | 0.8356 | 0.8319 |
| Vehicle | 0.6983 | 0.8566 | 0.8105 | 0.7843 |

Table 3: Comparison between LOOCV with Data Banzhaf and Data-OOB valuation frameworks for mislabeled detection (AUCROC) and weighted accuracy tasks.

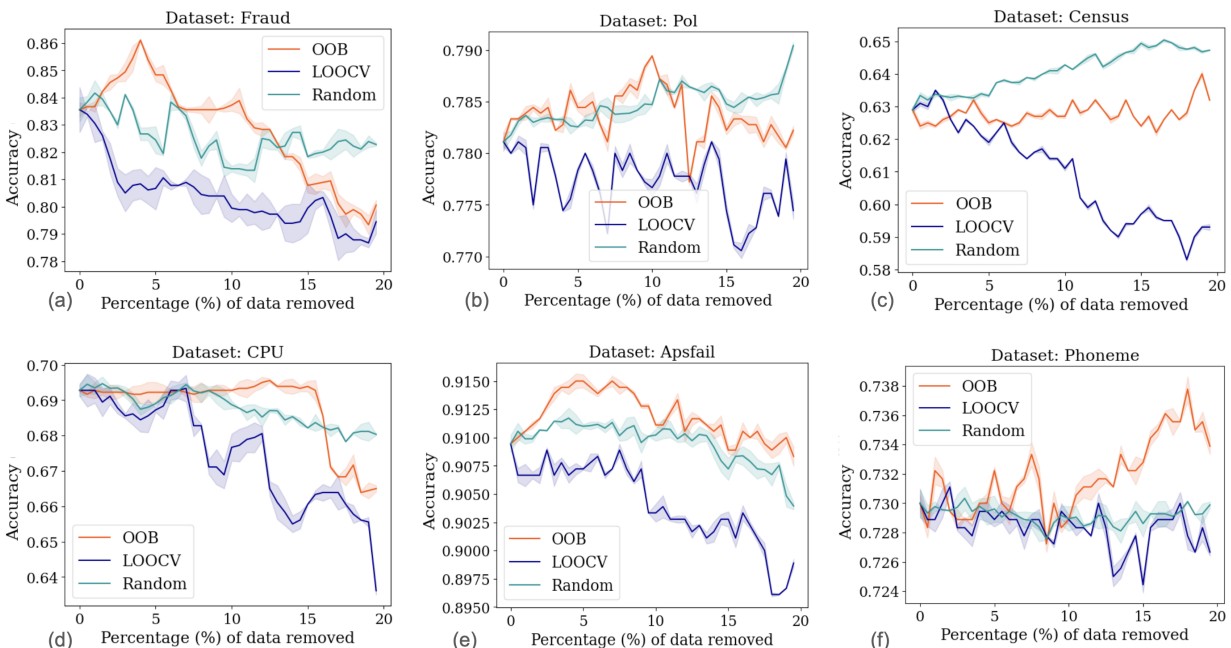

Figure 6: Data Removal Experiment- Test Accuracies of Logistic Regression models when data points are removed from the highest-valued, according to data values from LOOCV, Data-OOB and a Random Baseline.

setting. We consider experiments on three datasets- Adult Census data, Fraud Dataset and the Phoneme Dataset. These datasets were split into 8 datasets of size 50 each and the highest valued dataset was added first, following the experimental setup in Xu et al. (2021). The results from Figure 5 indicate that LOOCV performed better at this selection task. We include additional settings and setup in Appendix B.4.

**Comparison with Data-OOB valuation framework (Kwon & Zou, 2023):** Data-OOB (Out-Of-Bag) valuation framework proposes a valuation strategy that utilizes the out of bag error estimate from a bagging model (random forest). We conduct both mislabeled detection and weighted accuracy experiments on 7 datasets in Table 3. Our choice of label noise is 20% and we choose Data Banzhaf as the valuation framework for LOOCV. Data-OOB was proposed as a strong candidate for identifying outliers and noisy points present in the training set, and we find that it indeed performs better than LOOCV at assigning low values to these points. It is important to note that LOOCV based detection is quite strong on its own, as seen in Table 1. Interestingly, we find that LOOCV results in better weighted accuracies compared to Data-OOB, suggesting that LOOCV assigned higher weights to important points. In order to solidify this observation, we conduct a data removal experiment where a new logistic regression model is fit to the resulting dataset each

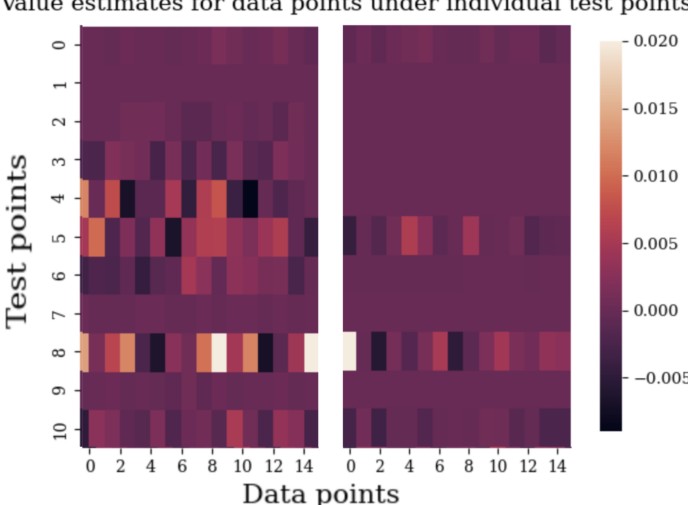

Figure 7: Heatmap of data values. *x-axis*: Data points with index 0-14 are valued using RLS (left) and logistic regression (right). *y-axis*: Individual test points.

time a data point is removed. We remove the highest valued data point first. It is expected that on removal of high-valued points the accuracy should decrease. We observe in Figure 6 that LOOCV performs better than Data-OOB in reducing the accuracy across all 6 datasets. Data-OOB performance is slightly worse than Random baseline in 3 of the datasets. This confirms that LOOCV is proficient in finding pivotal points, while also demonstrating a strong performance in detecting noisy points. We also conduct an ablation study for the data-removal experiment in Appendix B.5 where we assume clean label information and observe a similar trend in Figure 10. Additional settings and experimental setup is outline in Appendix B.5.

### 5.4 Why the choice of kernel regression is a powerful one?

We provide potential reasons for why RLS with Gaussian kernel is a strong proxy model for valuation. Logistic Regression is frequently used in past literature for model-based data valuation (Ghorbani & Zou, 2019; Kwon & Zou, 2021). Meanwhile, Kernel Regression is rarely used in this setting. Our values indicate strong performance by kernel regression, often superior (on noisy-label detection task) than values obtained from validation-based methods using Logistic Regression. We conduct a data valuation experiment on the Census Dataset using *individual test points* one at a time. We perform data valuation using Data Banzhaf framework for each test point using RLS and logistic regression. 15 training data points are selected at random from the whole set. We visualize a heatmap of their value estimates in Fig. 7. We observe that the values for each training point are much more sensitive for the RLS-based model than for their logistic regression-based equivalent. A possible reason for this behavior is the use of Gaussian Kernels in RLS which are more sensitive to localized change (or similarity) in the distance between two points - resulting in better detection of mislabeled samples as seen in Table 1. Meanwhile, models like logistic regression on average were not as sensitive to individual test points.

## 6 Conclusion & Limitations

In this work, we tackled the challenge of data valuation in the absence of a representative validation set. We examined the current validation-based data valuation techniques and identified their limitations. In response, we proposed a novel validation-free data valuation approach using LOOCV, and we propose to use RLS as proxy model. Our work offers a potential solution for scenarios where a clean, representative validation set is not available.

**Limitations.** Although we leverage RLS to make LOOCV computationally efficient, computing the exact Data Shapley (or other CGT-based approaches) is still computationally prohibitive as it requires computing LOOCV for all $2^N$ data subsets. An interesting future work is to explore potential learning algorithms where the exact Shapley value can be computed efficiently based on the performance scores from LOOCV (analogue to the famous KNN-Shapley (Jia et al., 2019a; Wang & Jia, 2023c)).

## Acknowledgements

RJ and the ReDS lab acknowledge support through grants from the Amazon-Virginia Tech Initiative for Efficient and Robust Machine Learning, the National Science Foundation under Grant No. IIS-2312794, NSF IIS-2313130, NSF OAC-2239622, and the Commonwealth Cyber Initiative. We are grateful to anonymous reviewers at TMLR for their valuable feedback.

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
