# A Error Analysis

For notation simplicity, we use $S \sim D, |S| = \ell$ to denote a random subsampling of a dataset $D$ of size $\ell$.

**Theorem 4** (Expected Error Bound for $\phi_{z_1}(U_{D_{\mathrm{val}}})$). *If both $U_{\mathcal{P}}$ and $U_{D_{\mathrm{val}}}$ have bounded range, then we have*

$$\mathbb{E}_{D \sim \mathcal{P}^N, D_{\mathrm{val}} \sim \mathcal{P}^k} \left[ |\phi_{z_1}(U_{D_{\mathrm{val}}}; D) - \phi_{z_1}(U_{\mathcal{P}}; D)| \right] \leq O\left( \frac{1}{\sqrt{k}} \right)$$

*Proof.* Since both $U_{\mathcal{P}}$ and $U_{D_{\mathrm{val}}}$ have bounded range, and since $U_{D_{\mathrm{val}}} = \mathrm{Perf}_{D_{\mathrm{val}}}(\mathcal{A}(S)) = \sum_{z \in D_{\mathrm{val}}} \mathrm{Perf}_{\{z\}}(\mathcal{A}(S))$, and $\mathbb{E}_{z \sim \mathcal{P}}[\mathrm{Perf}_{\{z\}}(\mathcal{A}(S))] = U_{\mathcal{P}}$, we have

$$\mathbb{E}_{D_{\mathrm{val}} \sim \mathcal{P}^k}[|U_{D_{\mathrm{val}}}(S) - U_{\mathcal{P}}(S)|] \leq \frac{C}{\sqrt{k}}$$

for some constant $C$ due to the standard sample error bound.

$$\mathbb{E}_{D \sim \mathcal{P}^N, D_{\mathrm{val}} \sim \mathcal{P}^k} \left[ |\phi_{z_1}(U_{D_{\mathrm{val}}}; D) - \phi_{z_1}(U_{\mathcal{P}}; D)| \right]$$

$$= \mathbb{E}_{D \sim \mathcal{P}^N, D_{\mathrm{val}} \sim \mathcal{P}^k} \left[ \left| \frac{1}{N} \sum_{\ell=0}^{N-1} \frac{1}{\binom{N-1}{\ell}} \sum_{S \subseteq D \setminus z_1, |S| = \ell} [((U_{D_{\mathrm{val}}}(S \cup z_1) - U_{\mathcal{P}}(S \cup z_1)) - (U_{D_{\mathrm{val}}}(S) - U_{\mathcal{P}}(S)))] \right| \right]$$

$$\leq \mathbb{E}_{D \sim \mathcal{P}^N, D_{\mathrm{val}} \sim \mathcal{P}^k} \left[ \frac{1}{N} \sum_{\ell=0}^{N-1} \frac{1}{\binom{N-1}{\ell}} \sum_{S \subseteq D \setminus z_1, |S| = \ell} [|U_{D_{\mathrm{val}}}(S \cup z_1) - U_{\mathcal{P}}(S \cup z_1)| + |U_{D_{\mathrm{val}}}(S) - U_{\mathcal{P}}(S)|] \right]$$

$$= \mathbb{E}_{D \sim \mathcal{P}^N} \left[ \frac{1}{N} \sum_{\ell=0}^{N-1} \mathbb{E}_{D_{\mathrm{val}} \sim \mathcal{P}^k, S \sim D \setminus z_1, |S| = \ell} [|U_{D_{\mathrm{val}}}(S \cup z_1) - U_{\mathcal{P}}(S \cup z_1)| + |U_{D_{\mathrm{val}}}(S) - U_{\mathcal{P}}(S)|] \right]$$

$$\leq \mathbb{E}_{D \sim \mathcal{P}^N} \left[ \frac{1}{N} \sum_{\ell=0}^{N-1} \mathbb{E}_{S \sim D \setminus z_1, |S| = \ell} \left[ \frac{C}{\sqrt{k}} + \frac{C}{\sqrt{k}} \right] \right]$$

$$= \frac{2C}{\sqrt{k}}$$

where the first inequality is due to triangle inequality. $\square$

**Theorem 5** (Expected Error Bound for $\phi_{z_1}(U_{\mathrm{LOOCV}})$). *If both $U_{\mathcal{P}}$ and $U_{\mathrm{LOOCV}}$ are average absolute prediction error and have a bounded range, then for linear regression models[1] , we have*

$$\mathbb{E}_{D \sim \mathcal{P}^N} \left[ |\phi_{z_1}(U_{\mathrm{LOOCV}}; D) - \phi_{z_1}(U_{\mathcal{P}}; D)| \right] \leq O\left( \frac{1}{\sqrt{N}} \right)$$

*Proof.* WLOG, assume the range of $U_{\mathcal{P}}$ and $U_{\mathrm{LOOCV}}$ are both $\leq 1$. By Tian et al. (2007), we know that for linear regression models we have

$$\mathbb{E}_{S \sim \mathcal{P}^m} \left[ |U_{\mathrm{LOOCV}}(S) - U_{\mathcal{P}}(S)| \right] \leq \frac{C}{\sqrt{m}}$$

for some constant $C$.

---

[1]Here, for the linear regression model $f(x) = \beta x$, the parameter $\beta$ is fit by solving the equation $\frac{1}{n} \sum_{i=1}^{n} X_i(Y_i - \beta^T X_i) = 0$. This equation has unique roots under mild assumptions outlined in Section 2 of Tian et al. (2007). While it is not clear how to extend the analysis to the least square training loss, several empirical studies have confirmed the moderate variance of generalization performance estimation of LOOCV (Tian et al., 2007; Zhang & Yang, 2015).

$$\mathbb{E}_{D \sim \mathcal{P}^N} \left[ |\phi_{z_1}(U_{\texttt{LOOCV}}; D) - \phi_{z_1}(U_{\mathcal{P}}; D)| \right]$$

$$= \mathbb{E}_{D \sim \mathcal{P}^N} \left[ \left| \frac{1}{N} \sum_{\ell=0}^{N-1} \frac{1}{\binom{N-1}{\ell}} \sum_{S \subseteq D \setminus z_1, |S|=\ell} [((U_{\texttt{LOOCV}}(S \cup z_1) - U_{\mathcal{P}}(S \cup z_1)) - (U_{\texttt{LOOCV}}(S) - U_{\mathcal{P}}(S)))] \right| \right]$$

$$\leq \mathbb{E}_{D \sim \mathcal{P}^N} \left[ \frac{1}{N} \sum_{\ell=0}^{N-1} \frac{1}{\binom{N-1}{\ell}} \sum_{S \subseteq D \setminus z_1, |S|=\ell} [|U_{\texttt{LOOCV}}(S \cup z_1) - U_{\mathcal{P}}(S \cup z_1)| + |U_{\texttt{LOOCV}}(S) - U_{\mathcal{P}}(S)|] \right]$$

$$= \mathbb{E}_{D \sim \mathcal{P}^N} \left[ \frac{1}{N} \sum_{\ell=0}^{N-1} \mathbb{E}_{S \sim D \setminus z_1, |S|=\ell} [|U_{\texttt{LOOCV}}(S \cup z_1) - U_{\mathcal{P}}(S \cup z_1)| + |U_{\texttt{LOOCV}}(S) - U_{\mathcal{P}}(S)|] \right]$$

$$= \frac{1}{N} \sum_{\ell=0}^{N-1} [\mathbb{E}_{S \sim \mathcal{P}^{\ell+1}} [|U_{\texttt{LOOCV}}(S) - U_{\mathcal{P}}(S)|] + \mathbb{E}_{S \sim \mathcal{P}^{\ell}} [|U_{\texttt{LOOCV}}(S) - U_{\mathcal{P}}(S)|]]$$

$$\leq \frac{1}{N} \left( 1 + C + \sum_{\ell=1}^{N-1} \left[ \frac{C}{\sqrt{\ell+1}} + \frac{C}{\sqrt{\ell}} \right] \right)$$

$$= O \left( \frac{1}{\sqrt{N}} \right)$$

$\square$

# B   Experiment settings and additional Experiment Details

## B.1   Experiment Settings for Table 1 and Table 2

**Datasets:** The evaluation deals with 9 datasets carefully picked from a wide variety of tasks such as image classification, speech recognition, value-based classification, etc. The selected datasets are popularly used for benchmarking in Data Valuation Literature. The datasets and their sources are mentioned in Table 4. Considering the computational bottleneck that accompanies Shapley-based subset sampling algorithms, we subsample a smaller dataset from the original training set. This subsampling is standard for all Shapley-based data valuation tasks. For CIFAR10 and MNIST, we consider a 2000-sized balanced subset for valuation tasks. For classification tasks Census and ApsFail Truck usage dataset, and speech recognition dataset Phoneme, we use a 1000-sized dataset. Finally, for OpenML datasets CPU, Fraud, Pol, and Vehicle we use a 200-sized subset. The selection of subset size is also affected by the complexity of the underlying problem. For instance, we find that Fraud dataset is easily separable with just 200 points (model achieves high accuracy).

**Models :** For our evaluation, LOOCV and baselines Self-Eval use a regularized least squares model (kernel regression) with a Gaussian Kernel ($\gamma = 1$) and regularization parameter ($\lambda = 0.1$). Our LOOCV implementation was helped a lot by the RL-score library on github (Pahikkala & Airola, 2016). For the validation-set based baseline, we use a logistic regression model since it achieves high accuracy on the standard classification task on the subsampled datasets. For CIFAR10 and MNIST, we additionally use a feature extractor (following the implementation in Ghorbani & Zou (2019), Jia et al. (2019a), etc.). For MNIST, we use the popular 2-layer CNN - LeNet (LeCun et al., 1998) as the feature extractor and use Adam as the learning algorithm (batch size is 32 and learning rate is 0.01). Similarly, for CIFAR10 we use a ResNet-18 architecture as the feature extractor with Adam as the learning algorithm (batch size is 200 and learning rate is 0.0006).

For the downstream task of weighted accuracy, we chose an independent model - Binary MLP with 2 Dense Layers for the non-image datasets. We use Adam as the learning algorithm with a batch size equal to the data size, and the learning rate is kept at 0.05. For CIFAR10 and MNIST, we keep the same model as the feature extractor mentioned above to compute the weighted accuracy. The only change is that here we set the batch size to be equal to 1000. Each result has been averaged over 50 times to eliminate stochasticity from Neural Network training.

**Data Valuation Settings :** In our evaluation we discuss two Shapley-based valuation frameworks namely Data Shapley (Ghorbani & Zou, 2019) and Data Banzhaf (Wang & Jia, 2023a). For Data Banzhaf, we use

| Dataset | Source |
|---------|--------|
| CIFAR10 | Krizhevsky et al. (2009) |
| MNIST | LeCun et al. (1989) |
| Census | Dua & Graff (2017) |
| Phoneme | https://www.openml.org/d/1489 |
| Apsfail | https://www.openml.org/d/41138 |
| CPU | https://www.openml.org/d/761 |
| Fraud | Dal Pozzolo et al. (2015) |
| Pol | https://www.openml.org/d/722 |
| Vehicle | Duarte & Hu (2004) |

Table 4: Datasets used in Table 1 and 2

the uniform sampling algorithm as used in the original paper. The valuation for size 200 (CPU, Fraud, Pol, Vehicle) and size 1000 datasets (Census, Phoneme, Apsfail) are computed with a total of 10,000 samples (total subsets), meanwhile the Banzhaf values for size 2000 datasets (MNIST and CIFAR10) are computed using a total of 30,000 samples. For Data Shapley, we use a permutation sampling (Monte-Carlo based) approach (Ghorbani & Zou, 2019). The total number of samples for size 200 datasets was 30,000. The number of samples for size 1000 datasets was 50,000 and for size 2000 datasets was 100,000 samples. In Table 5, for Beta Shapley we only deal with 200-sized (sub-sampled) datasets following the original implementation (Kwon & Zou, 2021).

**Data Valuation Frameworks Used:** We discuss the Data Valuation Frameworks used in context of our work. Data Shapley (Ghorbani & Zou, 2019) and Data Banzhaf (Wang & Jia, 2023a) were used in our Evaluation section, meanwhile Data Shapley and KNN-Shapley (Jia et al., 2019a) were used in the Motivation Section. We do not continue with KNN Shapley in the Evaluation section as it is not possible to extend it to validation-free setting. However, it is the state-of-the-art and hence we include it in the motivation section. Data Shapley was carefully defined in the main paper, and we define the others for better understanding here.

**KNN-Shapley.** Calculating the exact Shapley value can be computationally intensive, given that in general it necessitates computing $U(S)$ for every potential subset $S \subseteq D$. Yet, studies by (Jia et al., 2019a; Wang & Jia, 2023b) showed that for KNN, the exact Data Shapley score can be computed efficiently. Since its introduction, KNN-Shapley has quickly attracted research interest, inspiring many studies across various machine learning domains (Ghorbani et al., 2022; Liang et al., 2020; 2021; Courtnage & Smirnov, 2021). Typically, the utility of an unweighted KNN classifier is gauged by its validation accuracy. Given a validation set $D_{\text{val}} = \{z_i^{(\text{val})}\}_{i=1}^{N_{\text{val}}}$, the utility function of KNN, $U_{D_{\text{val}}}^{\text{KNN}}$, for a non-empty training subset $S$, is represented as $U_{D_{\text{val}}}^{\text{KNN}}(S) := \sum_{z^{(\text{val})} \in D_{\text{val}}} U_{z^{(\text{val})}}^{\text{KNN}}(S)$ where

$$U_{z^{(\text{val})}}^{\text{KNN}}(S) := \frac{1}{K} \sum_{j=1}^{\min(K,|S|)} \mathbb{1}\big[y_{\pi^{(S)}(j;x^{(\text{val})})} = y^{(\text{val})}\big] \tag{4}$$

captures the probability of a (soft-label) KNN-classifier accurately predicting the label for a validation data point $z^{(\text{val})} = (x^{(\text{val})}, y^{(\text{val})}) \in D_{\text{val}}$. $\pi^{(S)}(i; x^{(\text{val})})$ denotes the index of the $i$th nearest data point in $S$ relative to $x^{(\text{val})}$, using appropriate distance measures such as $\ell_2$ distance.

**Theorem 7** (Jia et al. (2019a))**.** *Consider the utility function in (4). Given a validation data point $z^{(val)} = (x^{(val)}, y^{(val)})$ and a distance metric $d(\cdot, \cdot)$, if we sort the training set $D = \{z_i = (x_i, y_i)\}_{i=1}^N$ according to $d(x_i, x^{(val)})$ in ascending order, then the Shapley value of each data point $\phi_{z_i}^{KNN}$ corresponding to utility*

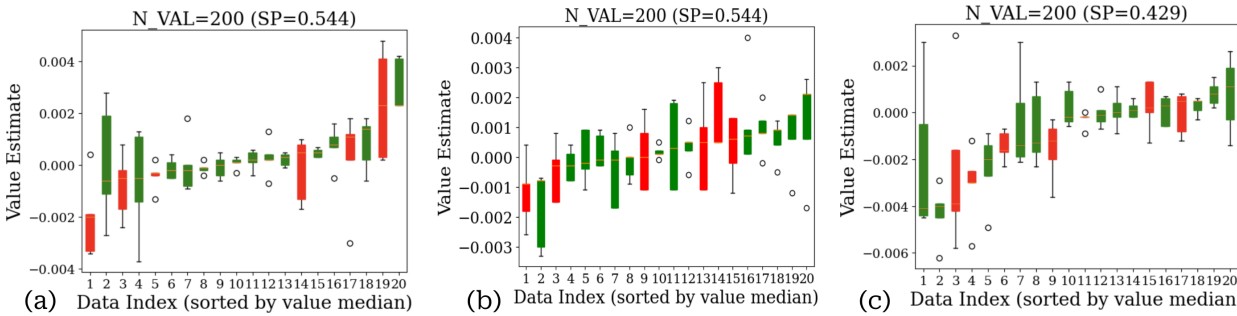

Figure 8: Box plots depict the data values(using Data Shapley) for 20 points randomly sampled from the Fraud (Fig (a)) and Phoneme (Fig (b)) Datasets over 5 different validation sets of equal size using Data Shapley. Fig c shows similar setup for Census Dataset using KNN Shapley framework. The mislabeled points are in red.

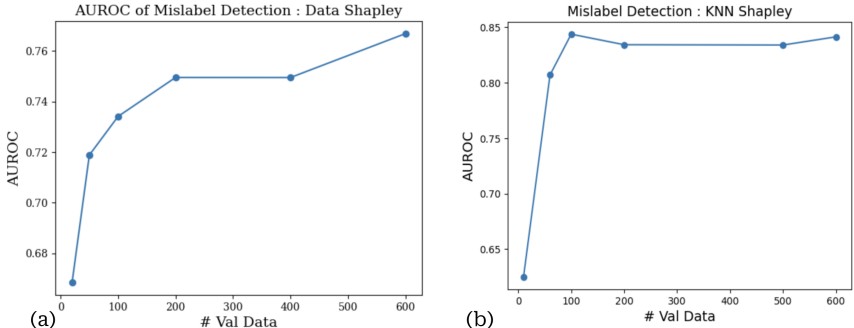

Figure 9: AUROC scores for mislabeled detection using (a) Data Shapley framework on Credit Card Dataset and (b) KNN Shapley framework on Census Dataset. Validation sizes were increased from 1% to 60% of the size of the training set.

*function $U_{z^{(val)}}^{KNN}$ can be computed recursively as follows:*

$$\phi_{z_N}^{KNN} = \frac{\mathbb{1}[y_N = y_{val}]}{\max(K, N)}$$

$$\phi_{z_i}^{KNN} = \phi_{z_{i+1}}^{KNN} + \frac{\mathbb{1}[y_i = y_{val}] - \mathbb{1}[y_{i+1} = y_{val}]}{K} \frac{\min(K, i)}{i}$$

As we can see, the exact $(\phi_{z_i}^{\text{KNN}})_{i=1}^N$ can be computed in $O(N \log N)$ runtime.

## B.2 Additional Motivation Experiments

In Figure 8, we provide additional experiments on the importance of choice of validation set. In Section 3, we observed that multiple same-sized validation sets drawn from the same distribution can cause variance in data values and therefore affect the ranking obtained in the Census Dataset. We show in this Figure that the same is true for Fraud and Phoneme Datasets (Fig 8 (a)-(b), with values for both clean and mislabeled points fluctuating between positive and negative values, and often taking on a high variance across the 5 different validation sets. The setup is identical to the one discussed in Section 3, and Data Shapley is the choice of valuation framework . In Fig 8 (c) we show that other valuation frameworks that rely on validation data may also suffer from the similar concern- by using KNN Shapley as the choice of valuation Framework on the Census Dataset- and we observe similar observations.

In Section 3, we also observed the effect of size of validation set on values obtained from validation-based data valuation frameworks (using KNN Shapley framework on Credit Card dataset (Yeh & Lien, 2009)). This dataset was chosen because noisy label detection on this dataset usually returns low AUROC scores (lower than possible on other datasets). We observed that as size of the validation set increased, the AUROC

| Dataset | LOOCV Beta (16,1) | LOOCV Beta (4,1) | Self-Eval Beta (16,1) | Self-Eval Beta (4,1) | Val Set Beta (16,1) | Val Set Beta (4,1) | Robust Volume |
|---|---|---|---|---|---|---|---|
| Phoneme | **0.8094** | 0.7831 | 0.7731 | 0.8550 | 0.5336 | 0.5692 | 0.4713 |
| Apsfail | 0.8189 | 0.7994 | 0.9155 | **0.9747** | 0.7325 | 0.8169 | – |
| Cpu | 0.8572 | **0.8886** | 0.8008 | 0.8392 | 0.7278 | 0.7397 | 0.6172 |
| Fraud | 0.8705 | 0.9139 | 0.9711 | **0.9786** | 0.6755 | 0.7753 | 0.5564 |
| Pol | 0.8133 | **0.8243** | 0.1738 | 0.2013 | 0.6925 | 0.83083 | 0.615 |
| Vehicle | **0.6599** | 0.6321 | 0.2739 | 0.3027 | 0.7178 | 0.7792 | – |

Table 5: Mislabled points detection: AUCs of noisy label detection rate are reported. This table studies application of the LOOCV paradigm to Beta Shapley and comparison with Self-Eval and Validation-based computation of Beta Shapley. All data sizes are 200. We highlight the best results of *validation-free approaches.*

scores of noisy label detection tasks increased - followed by a discussion on feasibility and availability of such a validation set. Rather, access to a limited validation set might imply compromising on quality of data values. We note in Fig 9 in two cases- Fig 9(a) shows similar effect on the Credit Card dataset using Data Shapley as well and Fig 9(b) attests similar observations on the Census Dataset using Data Shapley as our valuation framework. Our method (LOOCV) circumvents the requirement of a validation set and provides competitive AUROC scores, often superior to the validation-based counterparts.

**Settings for Fig 2 and Fig 8**: We sort the data points by their median. SP denotes the average Spearman's rank correlation over obtained data values.

### B.3 Comparison with Beta Shapley

**Beta Shapley.** By removing the efficiency axiom for the Shapley value, Kwon & Zou (2021) propose Beta Shapley as an alternative data valuation framework that enjoys an elegant mathematical formulation. Formally, given a utility function $U(\cdot)$ and a dataset $D = \{z_i\}_{i=1}^{N}$, the Beta Shapley value with hyperparameter $(\alpha, \beta)$ of a data point $z_i \in D$ is defined as

$$\phi_{z_i}^{\text{beta}}(U) := \sum_{k=1}^{n} \frac{w(k)}{n} \sum_{\substack{S \subseteq N \setminus \{i\}, \\ |S| = k-1}} [U(S \cup i) - U(S)]$$

where $w(k) := \frac{nB(k+\beta-1, n-k+\beta)}{B(\alpha, \beta)}$ and $B(\cdot, \cdot)$ is the Beta function.

In order to demonstrate LOOCV as a general paradigm applicable to Shapley-based methods, we also implement LOOCV for Beta Shapley (Kwon & Zou, 2021). We consider Beta (16,1) and Beta (4,1) which as shown by the authors shows better valuation performance than Beta (1,1) (which is essentially Data Shapley) for small data sizes. The reasoning behind this observation is that marginal contribution based on small data sizes have a large signal-to-noise ratio and such a distribution enables small marginal-contributions to have larger weight during valuation. Table 5 shows the application of Beta Shapley-based values to a noisy labeled detection task. We keep the validation size equal to the valuation dataset size which is 200. Note that Phoneme and Apsfail were 1000-sized in previous experiments but are 200-sized for this experiment

We find that LOOCV can perform well on the detection task and the quality of Beta Shapley values obtained is comparable to ones obtained with a validation set (superior for Phoneme and CPU datasets). We note that Self-Eval also performs really well and in fact gives the best performance for Apsfail and Fraud datasets. However we see a really poor performance for Pol and Vehicle datasets. We made the argument in Section 5.2 that self-eval values can be sporadic, even for some small datasets, and it is reinforced by the observations in Table 5. LOOCV does perform well for Pol and Vehicle datasets and hence is a more reliable option to obtain a validation-free data value, even for small data sizes.

### B.4 Comparison with Volume

We identify Volume (Xu et al., 2021) as another validation-free alternative to compute data values. However, volume is designed primarily for dataset valuation and not data-point valuation. We modify their implemen-

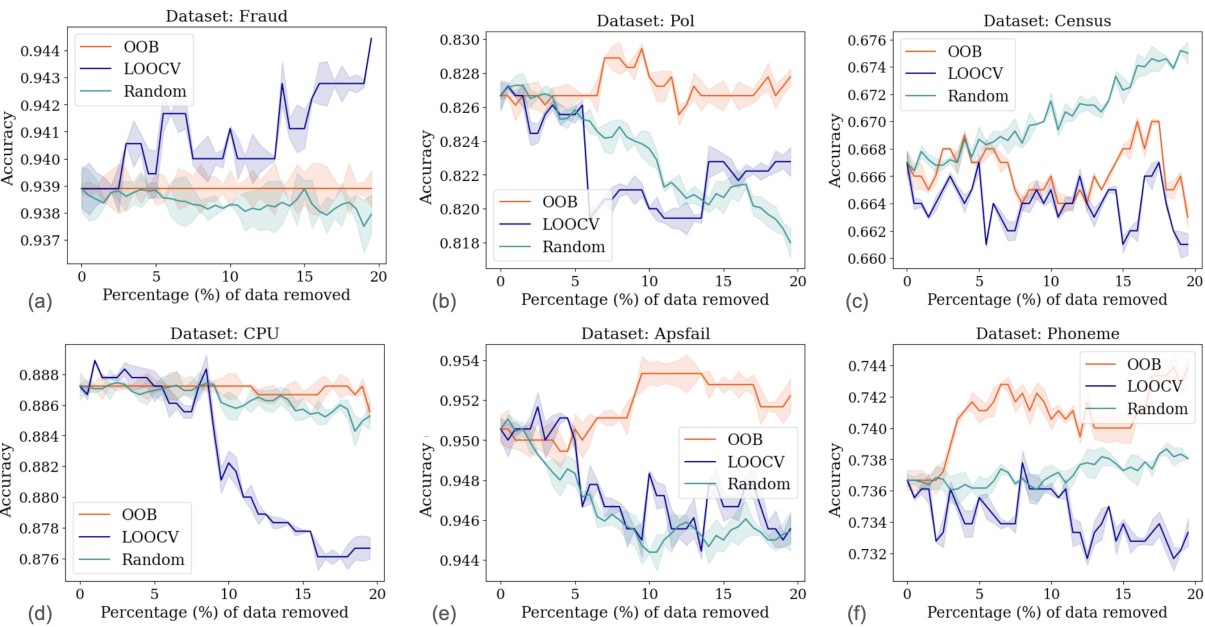

Figure 10: Data Removal Experiment- Test Accuracies of Logistic Regression models when data is removed. Data points are removed from the highest-valued according to data values from LOOCV, Data-OOB and a Random Baseline

tation to handle data valuation for small datasets (size 200). We test if Robust Volume can handle noisy labels to some extent. We attempt to see if a label-agnostic method can indeed detect 10 % noisy labels using AUC of detection rate. We see from Table 5 that Robust Volume is unable to perform well in the detection task, giving near-random performance in the detection task. This behavior is not unexpected since Volume has no label information. Volume-based valuation may not be ideal when the training data is not guaranteed to be clean. We note that volume-based values for Vehicle and Apsfail datasets are not added to the table because we ran into numerical computation issues (determinant exploded).

Additionally, we perform a dataset valuation experiment (Section 5.2) to compare if LOOCV can also perform reliable dataset valuation. We compare it to Robust Volume as a baseline. We chose the Phoneme, Census and Fraud Datasets for this experiment (Figure 5). The downstream task that we choose is data addition experiment - we add the highest valued dataset first and expect to see better or comparable performance to Volume since we are dealing with clean data. Each dataset was divided into 8 datasets each of size 50 each, and we use LOOCV and Volume to value these datasets. . Both methods use permutation sampling over 200 permutations to compute values. A validation set of 1000 points is used to determine performance and a Binary MLP with a learning rate of 0.05 is used as the model. We average over the validation accuracies over 10 runs. Our batch size was 32. The main observation is that our method can be comparable or even superior to Volume for dataset valuation.

### B.5 Comparison with Data-OOB

We compare our LOOCV valuation framework with Data-OOB (Kwon & Zou (2023)). While both methods provide a validation-free data valuation framework, they differ in their approach and applicability. LOOCV is a versatile framework, transforming conventional game-theoretic data valuation frameworks into validation-free alternatives. Data-OOB is a valuation strategy that relies on out-of-bag error from random forests to compute the marginal contribution of each data point.

In Section 5.3, we showed the respective strengths of Data-OOB and LOOCV. Data-OOB performed better on noisy label detection tasks, while LOOCV exhibited higher weighted accuracies when applied to value noisy datasets. In order to further compare the two label-free valuation strategies, we performed a Data Removal experiment on 6 datasets with 20 % mislabeled points. We observed that for each dataset, the

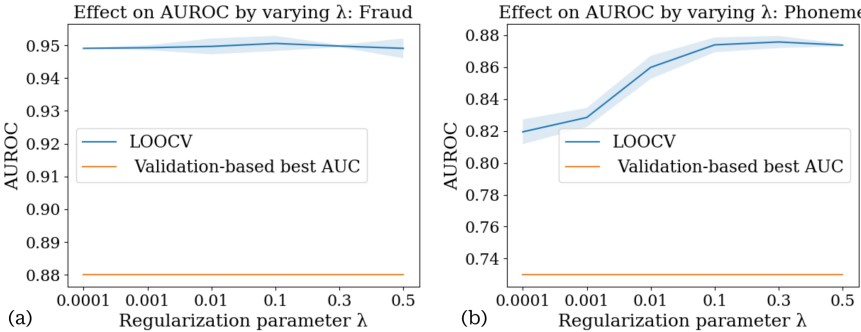

Figure 11: Ablation study experiment by varying regularization parameter $\lambda$ and observing the change in AUROC of mislabeled detection task. We observe that $\lambda$ of 0.1 is a safe choice since it results in high AUROC scores in both datasets.

accuracies reduced after removing data points with highest values for LOOCV, while the performance of Data-OOB was slightly better than random. We used Data Banzhaf as the choice of valuation framework for LOOCV in this study.

Furthermore, we perform another Data Removal experiment on the 6 datasets with the no mislabeled points (i.e. clean datasets). The objective behind this experiment is to further demonstrate that LOOCV is able to better assign pivotal points higher values. In Figure 10, we observe a similar trend of LOOCV reducing the accuracy after removing higher valued points (except on the Fraud dataset, where interestingly the peformance of both LOOCV and Data-OOB is near or worse than random removal of points). This study helped compare and highlight the differences between LOOCV and Data-OOB in their respective efficiencies and applicability.

### B.6 Ablation Study

The first ablation experiment performed is studying the effect of regularization parameter $\lambda$ and how to select it. We designed an experiment to observe the AUROC scores of noisy label detection performed by our LOOCV method using Data Banzhaf valuation framework. We chose two datasets for this setup-Phoneme(size 1000) and Fraud (size 200) (Table 4). We also include the best- AUC scores obtained using a Validation set as the baseline, to compare if the changes (if any) are better or worse than the baseline. We do this since the observation maintained in this paper remains that LOOCV is comparable or often superior to validation-based methods in noisy label detection task. We observe the results (Fig 11) in both cases (more noticable in case of Phoneme dataset) that for low values of $\lambda$ the AUROC scores drop below the AUROC scores achieved at higher values. We find that $\lambda$ of around 0.1 provides high AUROC scores in this task across both datasets. This is the value we use in our Evaluation section. We also note that despite the drop in AUROC scores, they remain above validation-based methods in case of both datasets.

The second ablation study compared the effect of varying the ratio of mislabeled examples present in the training set. The aim of this experiment is to observe if a high percentage of noisy labels in the training set affects the performance of our method. Since our method relies on training data for valuation, it is possible that higher noisy examples may reduce the utility of our method for this task. We choose Phoneme(size 1000) and Fraud datasets (size 200) for this ablation study and vary the mislabeled ratio from 1% - 20% (of the size of the training set). For a baseline, we provide the AUROC results from data values from validation-based valuation frameworks. We choose Data Banzhaf as the framework of choice. From Figure12 (a)-(b) - we observe that LOOCV values decrease as percentage of mislabeled points in Training set increase, as expected. We note that despite this behavior, our values stay consistently above the values from validation-based baseline. Self-eval seems to work well for size 200 Fraud Dataset but fails to perform well in case of size 1000 Phoneme dataset, reinforcing the points made in Figure 4 (a) that values obtained from Self-eval are not consistent as size of training dataset is increased. We conclude that LOOCV can provide quality values in the absence of a validation set despite change in noisy label ratio.

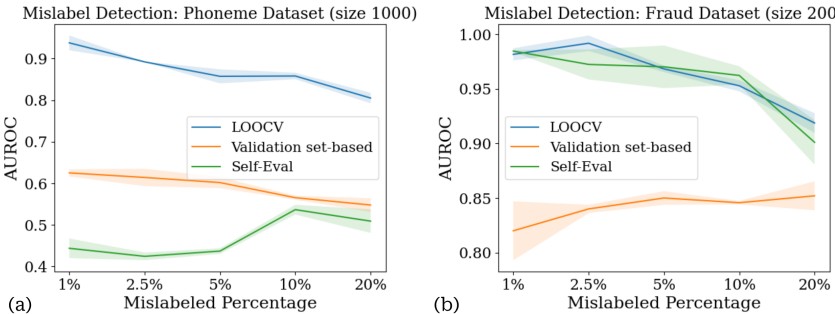

Figure 12: Ablation study experiment by varying noisy label ratio in Training Set of Phoneme and Fraud datasets and observing behavior across LOOCV, Self-Eval and validation-based data values using Data Banzhaf.

| Method | Runtime |
|---|---|
| LOOCV without retraining (our method) | 1.4276 sec |
| LOOCV via Kernel Regression Retraining | 12.885 sec |
| LOOCV via NN Retraining | 42.4590 sec |

Table 6: Comparison of runtimes of obtaining the LOOCV scores once using 1. Our Method 2. LOOCV with Kernel Regression retraining and 3. LOOCV with Neural Network (NN) retraining

**Note on noisy label detection performance :** We observed in Table 1 that LOOCV has achieved a superior performance in mislabeled detection task as can be observed by the AUC of mislabeled detection rate. The results for validation-based Data Banzhaf or Data Shapley may improve with a more appropriate choice of validation data or more number of sampled subsets (i.e. more permutations). However, it is difficult to determine what the best possible validation set might be and often it may not be feasible to achieve one either. Additionally, more permutations will imply added computational cost which is also not desirable. Therefore we want to note that the values obtained here are relative to the number of permutations used to compute the data value. We have used a fixed number of permutations for every same-sized dataset but performance may improve for certain datasets with increase in the number of permutations. But it is expected that LOOCV's performance will increase too.

It is worth mentioning that Self-Eval can also perform well for smaller sized (200) datasets (Table 2) . However, this performance was found to be sporadic - unlike LOOCV which performs well for each dataset.

### B.7 Computational Efficiency Experimental Study

**FLOPS Analysis.** In this analysis, we attempt to support the argument for computational efficiency benefit of our method- LOOCV with fast solution for Kernel Regression. For linear regression where $n$ is the number of training points and $p$ is the feature dimension, the naive approach to compute LOOCV requires retraining the model $n$ times, each time leaving out one data point, resulting in an overall complexity of $O(\max(n^2p^2, np^3))$. This is because each training iteration requires $O(\max(np^2, p^3))$ operations, and this process is repeated for all $n$ data points. In contrast, the hat matrix technique in LOOCV leverages precomputed values to avoid repeated model fitting. The hat matrix can be computed in $O(np^2 + p^3)$. Subsequently, adjustments for each left-out point can be made in $O(np)$ time. Thus, the total complexity of our approach is $O(np^2 + p^3)$, making it considerably more efficient than the naive method.

**Clock Time Results.** Besides the FLOPS analysis, we also provide a clock time comparison (Table 6 for the computation of LOOCV *once* on the Adult Census Dataset with size $n = 1000$ which has 102 dimensions, i.e., $p = 102$. We average runtimes over 5 different runs. We compare our method (LOOCV computation with proxy solution for Kernel Regression) with LOOCV computed via retraining using two methods, Kernel Regression retraining and Neural Network retraining (Binary MLP with a single layer). As

we can see from the table, our method is substantially faster than the naive retraining-based approaches for computing LOOCV.

**Note: Relationship between our method and Monte Carlo techniques.** We stress that our contribution here is to make a single utility function evaluation more efficient. We still rely on Monte Carlo estimators to estimate the Shapley value, and the overall runtime depends on the choice of estimators. The MC estimators' efficiency is orthogonal to our contribution, and we emphasize that our approach can be integrated with any MC estimators to achieve efficient Shapley value estimation.