# OpenReview forum: "Data Valuation in the Absence of a Reliable Validation Set"
_TMLR — Accepted by TMLR_

### Review · Reviewer_3AbG · 2024-06-14

**Summary Of Contributions:**

The authors claim that leave one out cross validation on the training set is better than a separate validation set.

**Audience:**

Yes

**Claims And Evidence:**

Yes

**Requested Changes:**

The authors need to add more motivations on when and how that the validation set hacking can happen, which seems to be the main contribution of the paper.

**Strengths And Weaknesses:**

Strengths:
1. The authors propose the problem of validation set hacking.
2. The authors did some experiments to support their claim.
3. The authors propose cv on a kernel regression model to speed up the process.

Weakness:
1. I feel that the authors propose many things people already knew. leave-one-out cross-validation/kernel regression are well-known techniques. The only contribution seems to be the problem validation set hacking.
2. For validation set hacking, the authors did not describe a specific situation when and how the hacking can happen.

---

> ### Author Response · Authors · 2024-08-01
>
> **Q1.Novelty**
>
> **A:** Our main contribution is not limited to validation set hacking, but rather providing a validation-free extension of traditional game-theoretic data valuation methods. Specifically, our contributions include
> - Identifying potential issues in validation-based frameworks, including size, quality, class-ratio, and gaming risks, and demonstrate their impact on data valuation through experiments (Fig 2 (a)-(b), Fig 3(a)-(b) in the main paper, and Fig 8 (a)-(c) and Fig 9(a)-(b) in the Supplementary material). These limitations can lead to inconsistent data values, compromised noisy label detection, and unfair payoffs to data providers, highlighting the need for a validation-free alternative for data valuation tasks.
> - Introducing a general paradigm for validation-free data valuation with LOOCV, using kernel regression as the surrogate model which enables efficient LOOCV computation, and providing error analysis.
>
> **Q2. Describing a specific situation when validation-set hacking can happen.**
>
> **A:** In the main paper (Section 3) we discussed a toy example to show that gaming valuation frameworks is a risk if a data contributor  has access to the validation set.
> In practice, in a data marketplace, a malicious seller can exploit the validation set by repeatedly querying slightly modified versions of their dataset and observing how the valuation changes. By using this feedback, they can employ zero-order optimization techniques to iteratively refine their dataset, effectively "gaming" the system to maximize their payoff. For instance, they might identify specific data points or features that significantly impact the valuation and manipulate them to inflate their dataset's value.

---

> > ### Comment · Reviewer_3AbG · 2024-08-01
> > **Thanks for the response**
> >
> > Thanks for the response. I think the contribution is interesting then. I changed my review.

---

### Review · Reviewer_PRvM · 2024-07-04

**Summary Of Contributions:**

The authors show that the choice of validation set can significantly impact the final data value scores. They convert a traditional validation-based game-theoretic data valuation method into a validation-free alternative. They utilize the cross-validation error as a surrogate for to evaluate the model's performance on a validation set. As computing the cross-validation error can be computationally expensive, they propose using the cross-validation error of a kernel regression model as an effective and efficient surrogate for the true performance score on the population.

**Audience:**

Yes

**Claims And Evidence:**

Yes

**Requested Changes:**

How do I see a trade-off between training size and validation size from the proposed error analysis?

**Strengths And Weaknesses:**

Strength: They compare the performance of the validation-free variant of existing data valuation techniques with their original validation-based counterparts. The results indicate that the validation-free variants generally match or often significantly surpass the performance of their validation-based counterparts.

Weakness: I could not see a trade-off between training size and validation size from the proposed error analysis.

---

> ### Author Response · Authors · 2024-08-01
>
> **Q1.Tradeoff between training and validation size?**
>
> **A:** The takeaway of Theorem 5, at least for linear regression, is that when the size of available validation set k is significantly less than N, LOOCV estimates of data value will be closer to the ground truth than values obtained from a utility function over that validation set, when the utility function is the (negative) mean square error. Therefore, only when the validation size k is of the order of N (e.g., 0.1*N) will the error be comparable with LOOCV. If the clean validation set is not large enough, then it is more favorable to use LOOCV.
>
> The error analysis shows that as the validation set size k increases, the data value scores from this validation set converge to values from the true population. Similarly when the training set size N increases, LOOCV data value scores converge to the true values of the populace. As we mention in Section 4.3 Theorem 5, at least for linear regression, when the size of available validation set k is significantly less than N, LOOCV estimates of data value will be closer to the ground truth than values obtained from a utility function over that validation set, when the utility function is the (negative) mean square error.

---

### Review · Reviewer_HH4j · 2024-07-17

**Summary Of Contributions:**

This paper focuses on data valuation and points out that the choice of validation data could greatly impact the final performance, potentially leading to limitations. To overcome this challenge, the authors propose a novel paradigm that transfers validation-dependent data valuation approaches into validation-free approaches. They also conduct a series of experiments on various choices, sizes, and class distributions to show the effectiveness of the proposed method.

**Audience:**

Yes

**Claims And Evidence:**

Yes

**Requested Changes:**

1. The results of the proposed method in Fig. 5 are kind of wired. The accuracy drops as the size increases. The authors should explain this and the metric (accuracy) is also not appropriate.
2. The computational complexity of the proposed method and other baselines should be added.

**Strengths And Weaknesses:**

Pro:
1. This paper is well-motivated. The authors start with a plain fact, i.e., the choice and quality of the validation set can significantly affect data values, which should be a major contribution of this paper. They conduct a series of experiments to indicate this and further propose their method.
2. The authors provide theoretical justification for the proposed method.

Con:
1. As mentioned in "Related work", Xu et al.'s work is quite relevant to this research. In my view, the validation accuracy in Fig. 5 is not appropriate here.
2. The proposed method utilizes cross-validation error and might be computationally expensive. The experiment lacks the comparison of computational efficiency with existing methods.

---

> ### Author Response · Authors · 2024-08-01
>
> **Q1 Choice of metric and explanations for the validation accuracy drop as the dataset becomes larger for Fraud in Figure 5?**
>
> **A:**
>  - In Figure 5, we design a dataset addition experiment, following the experimental designs in Xu et. al., converting real-life datasets into 8 datasets of the same size (50 each), and a large validation set from the population (whole dataset) of size 2000.
> - Specifically for the Fraud dataset, the accuracy drop seen as the dataset size increases is mainly due to the heterogeneous quality of the dataset.
> - Note that LOOCV is a solution to enable effective data valuation in the absence of a reliable validation set.  Data valuation will help determine how each dataset contributes towards downstream tasks, such as weighted training on their data values. Hence the choice of validation accuracy as our metric is an important one.
>
>
> **Q2. Computational efficiency?**
>
> **A:** Our validation-free data valuation paradigm contains two parts:
> (1) using Kernel regression’s LOOCV instead of validation performance as the utility function, and
> (2) using Monte Carlo-based algorithms to estimate the Shapley value with respect to the validation-free utility functions.
>
>
> - While LOOCV can be computationally expensive for general ML models, we propose using kernel regression as a proxy model. This approach has a key advantage: its LOOCV error can be computed without retraining the target neural networks for multiple times.
> - Computing the LOOCV error for kernel regression only requires training the model once on the input dataset. As a result, the runtime is similar to computing the kernel regression’s performance on the validation set.
> - Our approach also uses a Monte Carlo-based technique to estimate the Shapley value, similar to the original Data Shapley technique. The specific runtime complexity depends on the Monte Carlo algorithm used, which is separate from our main contribution.

---

> > ### Comment · Reviewer_HH4j · 2024-08-08
> > **Response to Authors**
> >
> > The response of authors raises my concerns.
> > 1. Do you mean that increasing the data size would lead to lower accuracy when the heterogeneous quality of dataset? I didn't get the point. Are these datasets under different data distribution?
> > 2. The authors mentioned computationally efficiency many times in this paper, while they did not provide detailed results (e.g., computational time and FLOPs) or even theoretical justification. Like they claimed that LOOCV is computationallu expensive and MC-based is not, but only claims.

---

> ### Author Response · Authors · 2024-08-09
>
> **Q1. Why increasing data size may lead to lower accuracy when the dataset is of heterogeneous quality?**
>
> **A:** We apologize for any confusion. To clarify, in our Figure 5, we randomly split the full dataset into subsets of 50 data points each. Due to the random nature of this splitting, **the quality of individual subsets can vary significantly.** As a result, not all data subsets contribute equally to the validation performance, and including low-quality subsets can potentially decrease model performance. In our experiment, we use data valuation algorithms to rank each subset and add them to the training data in order from highest to lowest value. By incorporating the highest-valued subsets first, we anticipate observing larger performance increases when the overall dataset is smaller. As we progress to adding lower-quality subsets, it becomes possible to observe a drop in model accuracy.
>
> This approach demonstrates the importance of data quality over quantity and highlights the potential of data valuation techniques in optimizing model training. By carefully selecting high-quality data subsets, we can improve model performance more efficiently than by simply increasing the size of training data.
>
> **Q2. Computational efficiency analysis & relationship with Monte Carlo estimators?**
>
> **A:**
> Thanks for the question! We love the suggestion of benchmarking the runtime. Here, we first clarify the baseline for efficiency comparison and present both the analysis and clock time results.
>
> Our paper first proposes a general framework which replaces the utility function being used in computing Data Shapley to the LOOCV, which avoids the dependency on the validation data.
>
> Computing this LOOCV for neural networks is computationally expensive since it involves retraining the model k number of times for a k-sized sample. Hence, we use kernel regression as the proxy model. This is a commonly used approximation in the literature [1, 2], which approximates the neural networks’ behavior asymptotically while reducing the computational burden for model retraining.
>
> However, naively computing LOOCV for kernel regression via retraining is still not efficient as shown in the FLOPS analysis below. Therefore, in our paper, we leverage the special property of kernel regression and significantly improve the runtime efficiency for computing LOOCV for kernel regression.
>
> - **FLOPS analysis.** For linear regression where $n$ is the number of training points and $p$ is the feature dimension, the naive approach to compute LOOCV requires retraining the model $n$ times, each time leaving out one data point, resulting in an overall complexity of $O(\max(n^2p^2, np^3))$. This is because each training iteration requires $O(\max(np^2, p^3))$ operations, and this process is repeated for all $n$ data points. In contrast, the hat matrix technique in our paper leverages precomputed values to avoid repeated model fitting. The hat matrix can be computed in $ O(np^2 + p^3) $. Subsequently, adjustments for each left-out point can be made in $ O(np) $ time. Thus, the total complexity of our approach is $ O(np^2 + p^3) $, making it considerably more efficient than the naive method.
>
> - **Clock time results.** Besides the FLOPS analysis, we also provide a clock time comparison for the computation of LOOCV once on the Adult Census Dataset with size n=1000 which has 102 dimensions i.e. p=102. As we can see from the table, our method is substantially faster than the naive retraining-based approaches for computing LOOCV.
> | Utility Function | Our method | LOOCV via Kernel Regression Retraining | LOOCV via Neural Network Retraining (Binary MLP with 1 hidden layer) |
> | --- | --- | --- | --- |
> | **Avg. Runtime** | 1.4276 sec | 12.885 sec | 42.4590 sec |
>
>
>
>
>
> **Relationship between our method and Monte Carlo techniques.** We stress that our contribution here is to make a single utility function evaluation more efficient. We still rely on Monte Carlo estimators to estimate the Shapley value, and the overall runtime depends on the choice of estimators. The MC estimators’ efficiency is orthogonal to our contribution, and we emphasize that our approach can be integrated with any MC estimators to achieve efficient Shapley value estimation.
>
> We will incorporate these additional results / discussion into the revised paper.
>
> [1] Zhaoxuan Wu, Yao Shu, and Bryan Kian Hsiang Low. Davinz: Data valuation using deep neural networks at initialization. ICML 2022.
> [2] Wang, Jingtan, et al. "Helpful or Harmful Data? Fine-tuning-free Shapley Attribution for Explaining Language Model Predictions." ICML 2024.

---

### Decision · Action_Editor_XvQw · 2024-08-23

**Recommendation:** Accept as is

**Comment:**

All reviewers agree with the observation that the problem this paper studies is important to the practice of modern machine learning, and that the contribution is significant - and I concur. The discussions between reviewers and authors have led to clarifications and to empirical evidence (in particular, those pertaining computational efficiency) that further supports the authors claims. I encourage the authors to incorporate these in their final, camera-ready version of this manuscript. There is a consensus that this paper is worthy of publication in TMLR, and all reviewers recommend a featured certification.

Finally, I encourage the authors to improve the quality of their figures (several of them are of poor resolution, and the aspect ratios of their text and numbers seem different from those in the main body of the manuscript).

**Audience:**

All reviewers agree that this contributions would be of interest to a large subgroup of the TMLR community, given the importance of the data valuation problem.

**Claims And Evidence:**

This paper studies the problem of data valuation, and in particular the issues that arise when a robust, high quality validation set is not available. As an alternative, the authors propose an validation data-free alternative based on LOOCV as utility function, as well as computationally efficient approximation to it. All reviewers support the observation that the claims made by the authors are well supported by the presented evidence, and that the contribution is significant, interesting, and clear.